# Priorities for intervention of childhood stunting in northeastern Ethiopia: A matched case-control study

**Sisay Eshete Tadesse**[1][☉], **Tefera Chane Mekonnen**[1][☉], **Metadel Adane**[iD][2][☉]*

**1** Department of Nutrition, School of Public Health, College of Medicine and Health Sciences, Wollo University, Dessie, Ethiopia, **2** Department of Environmental Health, College of Medicine and Health Sciences, Wollo University, Dessie, Ethiopia

☉ These authors contributed equally to this work.
* metadel.adane2@gmail.com

## Abstract

### Background

Stunting is a worldwide public health problem caused by factors that vary across regions, including in Ethiopia. Limited evidence to prevent stunting makes it difficult to design and prioritize appropriate interventions. Therefore, this study investigated the intervention priorities for the prevention of stunting among children 6–59 months old in Kemissie City Administration, northeastern Ethiopia.

### Methods

A community-based individual matched case-control study was conducted from January to April 2017 including 107 cases and 214 controls. Controls were selected and matched with cases using the matching variable of child's age. Data were collected by open data kit (ODK) software using a structured questionnaire. Data were analyzed using STATA version 13.0 and WHO (World Health Organization) Anthro 2005. A conditional logistic regression model was used for data analysis. From multivariable conditional logistic regression analysis, determinants of stunting were identified. A statistically significant level was declared by a conditional adjusted odds ratio (cAOR) with 95% confidence interval (CI) and *p*-value of less than 0.05.

### Main findings

The wealth index 52 (48.6%) of the cases and 108 (50.5%) controls were categorized as poor. The mean height-for-age z-score (HAZ) for the cases and controls was -2.79±.67 and -0.55±.92, respectively. One-sixth (16.8%) of the cases and 29 (13.6%) of the controls were given prelacteal feeding. A majority 82 (86.9%) of the cases and 137 (69.1%) of the controls had undernourished mothers/care-givers. Slightly less than one-third 35 (32.7%) of cases and one-fourth 53 (24.8%) of controls were affected by repeated episodes of diarrhea. Mother's body mass index (BMI) (conditional adjusted odds ratio [cAOR]) = 2.64; 95% CI: 1.28, 5.43), giving food priority to father (cAOR = 2.42; 95% CI: 1.23, 4.75), lack of exclusive

**Data Availability Statement:** All relevant data are within the manuscript and its Supporting Information files.

**Funding:** Wollo University funded this research. The funders had no role in study design, data collection and analysis, decisions to publish, interpretation of the data and preparation of the manuscript for publication.

**Competing interests:** The authors have declared that no competing interests exist.

**Abbreviations:** BMI, body mass index; CI, confidence interval; cCOR, conditional crude odds ratio; cAOR, conditional adjusted odds ratio; DDS, dietary diversity score; EBF, exclusive breast feeding; PCA, principal component analysis; WASH, water, sanitation, and hygiene; WHO, World Health Organization.

breastfeeding for at least 6 months (cAOR = 2.44; 95% CI: 1.15, 5.17), no intake of meat by child (cAOR = 2.35; 95% CI: 1.21, 4.58) and child having repeated diarrheal episodes (cAOR = 2.0: 95% CI: 1.07, 3.86) were factors associated with childhood stunting.

## Conclusion

Maternal nutritional status, food priority, duration of exclusive breastfeeding, no intake of meat and repeated episodes of diarrhea were the main determinants of stunting among children aged 6–59 months. Therefore, intervention measures to avert childhood stunting should include strengthening action on provision of essential nutrition, providing counseling to parents on giving food priority to children, promotion of optimal duration of breastfeeding and preventing diarrheal disease among children 6–59 months old.

## Introduction

Stunting is a worldwide public health problem affecting 155 million children under the age of five years. Africa is the only region with an increased absolute number of stunted children [1]. In a cumulative process, stunting can begin in the uterus and continue to about three years after birth. Over one-third of all deaths of those under age five are directly or indirectly associated with under-nutrition [2]. It has lifelong consequences, including delayed start of school, impaired cognitive and physical development, lower economic productivity, intergenerational effects, poorer reproductive performance and increased susceptibility to metabolic and cardiovascular diseases [2–4]. Stunting before the age of two years predicts poor cognitive and educational outcomes in later childhood and adolescence; its effect is irreversible after the age of two years [5, 6].

Stunting is associated with a number of long-term, interrelated factors, such as chronic insufficient protein and energy intake by children's, intrauterine growth retardation, lack of exclusive breastfeeding, inappropriate complementary feeding practices, frequent infection and food insecurity [7–10]. It is also associated with water, sanitation and hygiene, socio-demographic and economic factors [4].

Despite the efforts made by the government and stakeholders, stunting is still a serious public health problem affecting 38% of Ethiopian children under five. The annual target for reducing the occurrence of stunting has not been met. Poor progress towards the reduction of stunting might be due to lack of evidence that identifies key local determinants of stunting [11]. Previous studies from Ethiopia have revealed that determinants of stunting vary across regions and administrative towns in Ethiopia [12–15]. Studies done in this country have been mainly cross-sectional and include limited analytical studies [16], making them weak in generating evidence of the determinants of stunting [14, 15, 17, 18].

In the geographical area of this study, evidence to identify determinants of stunting among 6–59 months old children is limited or non-existent, which makes it difficult to design and prioritize appropriate interventions. Therefore, the aim of this study was to fill that gap by identifying the determinants of stunting using matched case-control study among children 6–59 months old in Kemissie City Administration, northeastern Ethiopia. This will help to design appropriate intervention measures for childhood stunting.

## Methods and materials

### Study area, design, period and study population

A community-based individual matched case-control study design was employed among children 6 to 59 months old from January to April 2017 in Kemissie City Administration, Oromia Special Zone, northeastern Ethiopia. Kemissie is located 325 km from Ethiopia's capital city Addis Ababa and has three urban and four rural *kebele*s (the smallest administrative units in Ethiopia). Based on reports of the Kemissie City Administration, the total population was 38,562 in 2017, of whom 18,895 (49.0%) were male and 19,667 (51.0%) were female. Furthermore, in 2015/16, there were 4,974 children between ages six and fifty-nine months, out these, 2,444 (49.1%) were male and 2,533 (50.9%) were female.

### Eligibility criteria

All children 6 to 59 months old with their mothers/care-givers who had been living in the selected *kebeles* for at least 6 months were included in the study. Children with visible congenital deformity and/or severe illness, and/or mothers/care-givers who were unable to communicate due to illness were excluded from the study.

### Sample size determination and sampling technique

The sample size was determined by a standard formula of matched case-control study [19] by considering the following assumptions: proportion of exclusive breast-feeding for a duration of less than or greater than six months among controls (*p*) is 29.66% [20], 95% CI (confidence interval) ($Z_{\alpha/2}$ = 1.96), 80% power ($Z_{\beta}$ = 0.84), odds ratio = 2,control to case ratio = 2, and non-response rate 10%. Therefore, the total number of study participants was 321; of these, 107 were cases and 214 were controls.

Among seven *kebeles* of the administrative city, three *kebeles* were randomly selected to be included in the study. A preliminary survey was conducted in the selected *kebeles* to identify cases and establish a sampling frame. The number of study participants was assigned for each selected *kebele* proportional to its population size.

**Selection of cases.** After establishing the sampling frame, cases were identified and selected during house-to-house transect walks in each selected *kebele* using a simple random sampling technique until the sample size was achieved.

**Selection of controls.** Controls were selected after the matching criterion of age was fulfilled according to other inclusion and exclusion criteria. Individual matching was carried out as one case followed by two controls based on three age categories from the same neighborhood found through transect walks, as explained above. Controls were matched to cases according to age: ±2 months for 6–11 months child, ±3 months for toddlers (12–23 months) and ±6 months for children (24–59) months [21]. The procedure ceased when the desired sample size was reached.

### Operational definition

**Cases.** Children aged 6 to 59 months whose length/height-for-age z-score was below -2SD (standard deviation) from the median height of the WHO (World Health Organization) reference population [22].

**Controls.** Children aged 6 to 59 months whose length/height-for-age z-score was -2SD and above from the median height of the WHO reference population [22].

## Data collection tools

Data were collected by open data kit software (ODK) through face-to-face interviews of mothers/care-givers using a structured questionnaire and observation checklist adapted from previous relevant literature [23, 24]. The collected data were sent to Kobo tool box common server, which was created before the start of data collection. Six BSc nurses as data collectors and one public health officer as a supervisor were recruited for data collection.

The questionnaire and observational checklist consisted of selected socioeconomic and demographic factors, health-care factors, child feeding practices and water, sanitation, and hygiene (WASH) related factors. Anthropometric measurements such as length/height of a child and height and weight of a mother were taken. For children 6–24 months old length was measured in a recumbent position to the nearest 0.1 cm using a standard lying board with a detachable sliding foot piece, while for children 25–59 months old and mothers, height was measured in a standing position to the nearest 0.1 cm using a standard vertical board with a detachable sliding head piece. Weight was measured to the nearest 0.1kg using the SECA Germany weight measurement scale.

Dietary Diversity Score (DDS) was calculated from single 24-hour dietary recall data. All the foods and liquids consumed the day before the study were categorized into seven food groups. Consumption of a food item from one of the groups was assigned a score of 1 and no consumption from that group was given a score of 0. Accordingly, a DDS of 7 points was computed by adding the values of all the groups. Then it was categorized as ≤4 and >4 [25].

## Data quality control

The data quality was assured through questionnaire translation to the local languages (Amharic) and back to English, conducting a pre-test of 5% of the sample size outside of the study area, selecting data collectors and supervisors based on their interest and experience in data collection, provision of two days training on the overall data collection process, checking and correcting of incomplete questionnaires by a supervisor on a daily basis and controlling of the overall data collection process by principal investigators. Additionally, two anthropometric measurements were taken by two data collectors and the average was taken. Calibration of the height and weight measurement scale was done at every five measurements. Outliers of the Anthropometric data were checked using the WHO Anthro 2005 software. The software detects an outlier if the z-score value is either <-5 or >5. The presence of outliers was checked by flag in WHO Anthro 2005 software. The result indicates that there was no outlier.

## Data management and analysis

Data were downloaded from the cloud server in csv. file extension format and exported to STATA version 13.0 for analysis. Data were cleaned by sorting and tabulating simple frequency tables. After dichotomizing the outcome variable into cases (1) and controls (0), descriptive statistics were computed and the result was reported using frequencies and percentages.

Wealth index was performed using principal component analysis (PCA). After checking the assumptions of PCA, rural and urban household asset from Ethiopian Demographic and Health Survey [26] were used to conduct the variables which were included in the PCA analysis. Radio, television, electric *mitad* (cooking grill), fridge, bed, roof made from corrugated iron sheet, walls made from cement, mobile, camel, ox, cow, goat, sheep, horse, *bajaj*, taxi, chair, table, sofa, lamp and shoes were variables included in the analysis. The summary components of the analysis were television, fridge, bed, roof made from corrugated iron sheet and wall made from cement. The total variance explained by the final component was 63.82%.

Then, wealth index was computed by calculating the mean and then the mean was divided in to three in order to categorize the study population as poor, medium and rich.

Anthropometric data were analyzed using the WHO Anthro 2005 software. The modeling strategy was based on the bi-variable (conditional crude odds ratio [cCOR]) and multi-variable (conditional adjusted odds ratio [cAOR]) conditional logistic regression analysis at 95% CI. From the bi-variable conditional logistic regression analysis, variables with $p<0.25$ were fitted into multivariable conditional logistic regression analysis in order to identify the determinants of stunting among children aged 6–59 months. Finally, variables with $p$-value$<0.05$ and adjusted odds ratio with 95% CI in the multivariable analysis were taken as statistically significant.

### Ethical considerations

The proposal to conduct the research was approved by the ethical review board of the College of Medicine and Health Sciences of Wollo University, and an ethical approval letter was obtained. A support letter was obtained from Kemissie City Administration health office for the selected *kebeles* in order to obtain approval to conduct the study. After the purpose and objective of the study and the voluntary nature of participation was explained to the mothers/care-givers of participating children, written informed assent and consent were obtained from the mothers/caregivers of both cases and controls, assent on behalf of the participating children aged 6–59 months, and consent for the mothers/caregivers themselves. All the information was kept confidential, and no individual identifiers were used.

## Results

### Socio demographic and economic characteristics

Three hundred twenty-one (321) mothers/care-givers with children 6–59 months old, made up of 107 cases and 214 controls, were recruited in this study. The mean age of the cases and controls was 29.97±13.70 and 30.36±14.74 months, respectively. Nearly one-third 67 (62.6%) of the cases and 103 (48.1%) of the controls were male. The mean height-for-age z-score (HAZ) for the cases and controls was -2.79±.67 and -0.55±.92, respectively. Based on the result of principal component analysis of wealth index 52 (48.6%) of the cases and 108 (50.5%) the controls were categorized as lowest category or poor (Table 1).

### Child feeding practices and maternal nutritional status

A majority 91 (85.0%) of the cases and 197 (92.0%) of the controls were breastfed. Slightly less than two-thirds 68 (63.6%) of the cases and 178 (83.2%) of the controls were fed breast milk within one hour after birth. With respect to prelacteal feeding practices, 18 (16.8%) of the cases and 29 (13.6%) of the controls were given prelacteal feeding. Nearly one-fourth 26 (24.3%) of the cases and 23 (10.7%) controls were not breastfed exclusively for the first six months of life.

Forty-five (42.1%) of the cases and 65 (30.4%) of the controls were breastfed less than 8 times per day and more than half 60 (56.0%) of the cases and 122 (57.0%) of the controls were started complementary feeding at six months of age. The number of mothers/care-givers with a BMI of greater than or equal to 18.5 for cases were 14 (13.1%) and for controls 64 (29.9%). With respect to food priority, 39 (36.5%) of the cases and 42 (19.6%) of the controls gave priority to the fathers. A majority 82 (86.9%) of the cases and 137 (69.1%) of the controls had undernourished mothers/care-givers (Table 2).

Table 1. Socio-demographic characteristics of children aged 6–59 months in Kemissie City Administration, northeastern Ethiopia, January to April 2017.

| Variables | Category | Cases (N = 107) | Controls (N = 214) |
|---|---|---|---|
| | | n (%) | n (%) |
| Sex of child | Male | 67 (62.6) | 103 (48.1) |
| | Female | 40 (37.4) | 111 (51.9) |
| Marital status of mothers/care-givers | Married | 104 (97.2) | 198 (92.5) |
| | Other* | 3 (2.8) | 16 (7.5) |
| Age of mothers/care-giver (years) | 20–24 | 45 (42.1) | 52 (24.3) |
| | 25–29 | 32 (29.9) | 83 (38.8) |
| | 30–34 | 16 (14.9) | 49 (22.9) |
| | 35+ | 14 (13.1) | 30 (14.0) |
| Educational status of mothers/care-givers | Illiterate | 65 (60.8) | 110 (51.4) |
| | Elementary | 36 (33.6) | 76 (35.5) |
| | High school and above | 6 (5.6) | 28 (13.1) |
| Occupation of mothers/care-givers | Housewife | 101 (94.3) | 192 (89.7) |
| | Other | 6 (5.7) | 19 (10.3) |
| Wealth index | Poor | 52 (48.6) | 108 (50.5) |
| | Medium | 21 (19.6) | 46 (21.5) |
| | Rich | 34 (31.8) | 60 (28.0) |
| Household size (persons) | ≤3 | 23 (30.8) | 48 (22.4) |
| | 4–5 | 45 (42.1) | 82 (38.3) |
| | ≥6 | 29 (27.1) | 84 (39.3) |

*widowed and divorced.

### Health-care related factors

One-third 36 (33.6%) of the cases and 44 (20.5%) of the controls were born at home. Only 34 (31.8%) of the cases' and 91 (42.5%) of the controls' mothers/care-givers obtained post-natal care, while the rest did not get postnatal care. Slightly less than one-third 35 (32.7%) of cases and one-fourth 53 (24.8%) of controls were affected by repeated episodes of diarrhea (Table 3).

### Water, sanitation, and hygiene (WASH) related factors

Almost all 105 (98.1%) of case households and 201 (98.1%) of control households had a latrine. Mothers/care-givers practiced hand washing at critical times in 100 (93.5%) of case households and 185 (86.4%) of control households. Most of the cases 91 (85.1%) and 180 (84.1%) of the controls used tap/public water for drinking purposes. In forty-seven (43.9%) of the case and 65 (30.3%) of the control households, solid waste was disposed of in the open fields. The rest of the households managed solid waste through burying, burning or collection by municipalities.

### Determinants of stunting from multivariable conditional logistic regression analysis

The multivariable conditional logistic regression model revealed that children who were born to mothers of low BMI were almost 2.6 times more likely to become stunted as compared to those children who were born to mothers of normal BMI (cAOR = 2.64; 95% CI: 1.28, 5.43). Those children who were not exclusively breastfed up to six months of age were 2.4 times as likely to be stunted as those who were exclusively breastfed until six months of age (cAOR = 2.44; 95% CI: 1.15, 5.17) (Table 4).

**Table 2.  Practices for feeding children 6–59 months old in Kemissie City Administration, northeastern Ethiopia, January to April 2017.**

| Variables | Category | Cases (N = 107) | Controls (N = 214) |
|---|---|---|---|
| | | n (%) | n (%) |
| Initiation of breastfeeding after birth | Within 1 hour | 68 (63.6) | 178 (83.2) |
| | 1–3 hours | 13 (12.1) | 22 (10.3) |
| | >3 hours | 26 (24.3) | 14 (6.5) |
| Prelacteal feeding | Yes | 18 (16.8) | 29 (13.6) |
| | No | 89 (83.2) | 185 (86.4) |
| Duration of exclusive breastfeeding (EBF) | Six months | 81 (75.7) | 191 (89.3) |
| | < or > 6 months | 26 (24.3) | 23 (10.7) |
| Frequency of breastfeeding (BF) | Suboptimal | 45 (42.1) | 65 (30.4) |
| | Optimal | 62 (57.9) | 149 (69.6) |
| Initiation of complementary feeding | Timely | 60 (56.0) | 122 (57.0) |
| | Early/late | 47 (44.0) | 92 (43.0) |
| Consumption of diary and dairy products | Daily | 88 (82.2) | 190 (88.8) |
| | 1–2 times/week | 13 (12.2) | 10 (4.7) |
| | Not consumed | 6 (5.6) | 14 (6.5) |
| Consumption of eggs | 1–2 times/week | 64 (59.8) | 152 (71.0) |
| | Once/two weeks | 13 (6.2) | 26 (12.2) |
| | Not consumed | 30 (28.0) | 36 (16.8) |
| Consumption of Vitamin A-rich fruits and vegetables | Every other day | 13 (12.1) | 42 (19.6) |
| | 1–2 times/week | 26 (24.3) | 63 (29.4) |
| | Once/two weeks | 8 (7.5) | 22 (10.3) |
| | Not consumed | 60 (56.1) | 87 (40.7) |
| Consumption of meat | Daily | 23 (21.4) | 72 (33.7) |
| | 1–2 times/week | 16 (15.0) | 39 (18.2) |
| | Not consumed | 68 (63.6) | 103 (48.1) |
| Dietary diversity score (DDS) | ≤4 | 94 (87.9) | 169 (79.0) |
| | >4 | 13 (12.1) | 45 (21.0) |
| Food priority | Father | 39 (36.5) | 42 (19.6) |
| | Child | 31 (28.9) | 92 (43.0) |
| | All equal | 37 (34.6) | 80 (37.4) |
| BMI of mothers/care-givers | ≥18.5 | 14 (13.1) | 64 (29.9) |
| | < 18.5 | 82 (86.9) | 137 (69.1) |

The odds of being stunted among children who had repeated episodes of diarrhea were 2 times higher than their counterparts who did not had repeated episodes of diarrhea (cAOR = 2.0: 95% CI: 1.07, 3.86). The odds of being stunted among children who did not get meat per two weeks were almost 2.4 times higher than among those who ate meat daily (cAOR = 2.35; 95% CI: 1.21, 4.58). Children from households where food priority was given to fathers were 2.4 times as likely to be stunted as those where food priority was given to the children (cAOR = 2.42; 95% CI: 1.23, 4.75) (Table 4).

## Discussion

Identifying factors associated with stunting is a first step to design appropriate intervention measures to tackle the problem and its consequences in Kemissie City Administration in particular and in Ethiopia as a whole. Therefore, this study aimed to identify the determinants of stunting among children using individual matched case-control study design. Overall,

**Table 3. Healthcare-related factors among children 6–59 months old in Kemissie City Administration, northeastern Ethiopia, January to April 2017.**

| Variables | Category | Cases (N = 107) | Controls (N = 214) |
|---|---|---|---|
| | | n (%) | n (%) |
| Number of antenatal visits | <2 times | 14 (13.1) | 20 (9.4) |
| | 2–4 times | 91 (85.0) | 182 (85.0) |
| | >4 times | 2 (1.9) | 12 (5.6) |
| Place of birth | Home | 36 (33.6) | 44 (20.5) |
| | Health facility | 71 (66.4) | 170 (79.5) |
| Birth weight of child | <2.5 kg | 41 (38.3) | 62 (28.9) |
| | 2.5–4 kg | 55 (51.4) | 136 (63.6) |
| | >4 kg | 10 (9.3) | 16 (7.5) |
| Postnatal care (PNC) | Yes | 34 (31.8) | 91 (42.5) |
| | No | 73 (68.2) | 123 (57.5) |
| Type of illness within the previous two weeks | Not ill | 54 (50.5) | 134 (62.6) |
| | Diarrhea | 35 (32.7) | 53 (24.8) |
| | Respiratory infection | 18 (16.8) | 27 (12.6) |
| Immunization | Yes | 103 (96.3) | 108 (97.2) |
| | No | 4 (3.7) | 6 (2.8) |

**Table 4. Determinants of stunting among children 6–59 months old in Kemissie City Administration, Oromia special zone, northeastern Ethiopia, January to April 2017.**

| Variables | Category | Stunting | | cCOR (95% CI) | cAOR (95% CI) |
|---|---|---|---|---|---|
| | | Cases | Controls | | |
| | | (N = 107) | (N = 214) | | |
| | | n (%) | n (%) | | |
| BMI of mothers/care-givers | < 18.5 | 82 (86.9) | 137 (69.1) | 2.57 (1.38, 4.81) | 2.64 (1.28, 5.43) |
| | ≥18.5 | 14 (13.1) | 64 (29.9) | 1 | 1 |
| Duration of EBF | < or > 6 months | 26 (24.3) | 23 (10.7) | 2.73 (1.44, 5.16) | 2.44 (1.15, 5.17) |
| | 6 months | 81 (75.7) | 191 (89.3) | 1 | 1 |
| Education status of mothers/care-givers | Illiterate | 65 (60.8) | 110 (51.4) | 2.76 (1.08, 7.01) | 2.36 (0.81, 6.88) |
| | Elementary | 36 (33.6) | 76 (35.5) | 2.21 (0.81, 5.81) | 2.18 (0.73, 5.75) |
| | High school and above | 6 (5.6) | 28 (13.1) | 1 | 1 |
| Meat | 1–2 times/week | 16 (14.9) | 39 (18.2) | 1.25 (0.60, 2.58) | 1.57 (.62, 4.00) |
| | None | 68 (63.6) | 103 (48.2) | 1.96 (1.14, 3.37) | 2.35 (1.21, 4.58) |
| | Daily | 23 (21.5) | 72 (33.6) | 1 | 1 |
| Type of illness | Respiratory infection | 19 (17.8) | 27 (12.6) | 1.76 (0.89, 3.33) | 2.05 (0.94, 4.47) |
| | Diarrhea | 34 (31.8) | 53 (24.8) | 1.61 (0.93, 2.78) | 2.00 (1.07, 3.86) |
| | Not ill | 54 (50.5) | 134 (62.6) | 1 | 1 |
| Food priority | Father | 39 (36.5) | 42 (19.6) | 2.96 (1.59, 5.54) | 2.42 (1.23, 4.75) |
| | Children | 31 (28.9) | 92 (43.0) | 1 | 1 |
| | All equal | 37 (34.6) | 80 (37.4) | 1.31 (0.73, 2.34) | 1.29 (0.65, 2.55) |
| Wealth Index | Poor | 52 (48.6) | 108 (50.5) | 0.85 (0.49, 1.45) | |
| | Medium | 21 (19.6) | 46 (21.5) | 0.81 (0.41, 1.57) | |
| | Rich | 34 (31.8) | 60 (28.0) | 1 | |
| Water source | Protected | 102 (95.3) | 196 (91.6) | 0.42 (0.14, 1.28) | 0.36 (0.11, 1.18) |
| | Unprotected | 5 (4.7) | 18 (8.4) | 1 | 1 |

[1]Reference category; cCOR, conditional crude odds ratio; cAOR, conditional adjusted odds ratio; EBF, exclusive breast feeding; BMI, body mass index.

nutritional status of mother/care-givers, household food priority, repeated episodes of diarrhea, meat consumption and postnatal care were factors significantly associated with stunting among children 6–59 months old.

Among the identified factors of stunting, maternal nutritional status (chronic energy deficiency) appeared to be negatively associated with stunting among children 6–59 months old. Those children who were born to mothers of low BMI were almost 2.6 times more likely to become stunted as compared to those children who were born to mothers of normal BMI. This finding is in line with previous studies, which stated that poor nutritional status of mothers before conception and during pregnancy is associated with giving birth to stunted children [20, 27]. This might be because inadequate nutrition during pregnancy is associated with insufficient nutrient transfer to the fetus, which accounts for a large proportion of growth retardation in the fetal environment. This would result in a decreased birth weight and growth impairment of the child in later life.

Those children who were not exclusively breastfed up to six months of age were 2.4 times as likely to be stunted as those children who exclusively breastfed until six months of age. Other literature revealed a similar significantly higher risk of being stunted among children who were not exclusively breastfed [16, 28–30]. This may be because the short period of breastfeeding is insufficient to provide adequate dietary intake of energy, protein and micronutrients for optimal mental and physical growth [31]. This condition may lead to faltering growth during the childhood period. Additionally, it may also be associated with poor hygienic conditions, which can cause malnutrition through infection. Extending exclusive breastfeeding beyond six months of age alone did not provide adequate nutrients for the fast-growing babies. This finding supports a study that stated that stunting mostly occurs in the first 6–18 months of life [32].

The odds of being stunted among children who had repeated episodes of diarrhea were approximately twice as high as those who did not. This finding is consistent with studies done in other parts of the world [18, 30, 33, 34]. This could be due to the effect of diarrheal infection on appetite, nutrient absorption, increment of metabolic requirements and increasing nutrient loss [2]. It might also be due to the direct relationship between diarrhea and malnutrition; that is, diarrhea can lead to malnutrition and malnutrition can predispose for diarrhea. Some studies indicated that, for urban under-five-year-old children, diarrhea was caused by poor WASH status [35–37], which in turn might also have caused these children's malnutrition. According to the UNICEF conceptual framework, diarrhea is an immediate cause of malnutrition.

An important association was also observed between intake of meat and the occurrence of stunting. The odds of being stunted among children who did not get meat per two weeks were almost 2.4 times as high as among those children who ate meat on a daily basis. Other similar studies revealed that children's intake of protein from animal sources was associated with reduced odds of being stunted [34, 38]. This could be because foods of animal origin are good sources of easily digestible protein that has higher conversion and absorption rates by the body compared to plant-based protein. Thus, consistent intake of protein from animal sources is essential for optimal physical growth and a healthy future life. This finding is also supported by a study conducted on diets in Africa that stated that having staple foods with low protein content such as cassava and rice could be a risk factor for poor childhood development [39].

Another main finding of the current study is that children of households where food priority was given to fathers were 2.4 times as likely to be stunted compared to those where food priority was given to the children. This finding is similar to that of a study done in the Medebay Zana District, Tigray, Ethiopia [14]. This could be due to a family's inability to meet the increasing nutritional needs of a growing baby as a result of inadequate access to food.

Consumption of foods that do not meet the child's nutrient needs in terms of the amount and composition are a direct cause of stunting [40].

Since the study design was retrospective in nature, it may have been affected by recall bias; and the usual intake of nutrients may also be affected by seasonal variation. Generalization of the findings to the national level is also impossible due to the small setting of this study.

## Conclusion

This study revealed that chronic energy deficiency of mothers/care-givers (BMI<18.5 kg/m$^2$), giving food priority to father, inappropriate duration of breast-feeding practices, inadequate consumption of meat and repeated episodes of diarrhea were factors associated with stunting. Therefore, to avert stunting among children aged 6–59 months, intervention priorities should include strengthening the implementation of essential nutrition action, such as improving maternal nutritional status during pregnancy, increasing compliance of iron and folic acid intake and promoting optimal child feeding practices within the first 1,000 days of life). Additionally, designing intervention measures should focus on counseling of family members to provide food priority for their children and to prevent diarrheal disease among children 6–59 months old.

Unless we intervene on stunting, the country will be unable to achieve the UN sustainable development goals by 2030, particularly Goal 3, target 3.2 focused on ending preventable deaths of newborns and reduce under-five child mortality to 25 per 1,000 live births or lower. Due to its adverse effect on the mental development, childhood stunting hinders the overall development of a country. Finally, researchers are recommended to conduct further research on stunting using follow-up studies that consider the seasonal effect.

## Supporting information

**S1 File. Household survey questionnaire in English version.**
(DOCX)

**S2 File. Household survey questionnaire in Amharic version.**
(DOCX)

## Acknowledgments

We would like to acknowledge Kemissie City Administration and health officials in each studied area *kebele* for the support we received during the data collection. We thank mothers/care-givers of the studied children for their dedication and commitment in providing information during the survey. We also thank data collectors and supervisors for their cooperation during the process of data collection.

## Author Contributions

**Conceptualization:** Sisay Eshete Tadesse, Tefera Chane Mekonnen, Metadel Adane.

**Data curation:** Sisay Eshete Tadesse, Tefera Chane Mekonnen, Metadel Adane.

**Formal analysis:** Sisay Eshete Tadesse, Tefera Chane Mekonnen, Metadel Adane.

**Funding acquisition:** Sisay Eshete Tadesse, Tefera Chane Mekonnen, Metadel Adane.

**Investigation:** Sisay Eshete Tadesse, Tefera Chane Mekonnen, Metadel Adane.

**Methodology:** Sisay Eshete Tadesse, Tefera Chane Mekonnen, Metadel Adane.

**Project administration:** Sisay Eshete Tadesse, Tefera Chane Mekonnen, Metadel Adane.

**Resources:** Sisay Eshete Tadesse, Tefera Chane Mekonnen, Metadel Adane.

**Software:** Sisay Eshete Tadesse, Tefera Chane Mekonnen, Metadel Adane.

**Supervision:** Sisay Eshete Tadesse, Tefera Chane Mekonnen, Metadel Adane.

**Validation:** Sisay Eshete Tadesse, Tefera Chane Mekonnen, Metadel Adane.

**Visualization:** Sisay Eshete Tadesse, Tefera Chane Mekonnen, Metadel Adane.

**Writing – original draft:** Sisay Eshete Tadesse, Tefera Chane Mekonnen, Metadel Adane.

**Writing – review & editing:** Sisay Eshete Tadesse, Metadel Adane.

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
