## [Decision Letter · Decision Letter 0]

21 Apr 2020

PONE-D-19-23907

Determinants of stunting among children aged 6-59 months in Kemissie City Administration, North Eastern Ethiopia: A community-based matched case-control study

PLOS ONE

Dear Dr Adane (PhD),

Thank you for submitting your manuscript to PLOS ONE. After careful consideration, we feel that it has merit but does not fully meet PLOS ONE’s publication criteria as it currently stands. Therefore, we invite you to submit a revised version of the manuscript that addresses the points raised during the review process.

We would appreciate receiving your revised manuscript by Jun 05 2020 11:59PM. To enhance the reproducibility of your results, we recommend that if applicable you deposit your laboratory protocols in protocols.io, where a protocol can be assigned its own identifier (DOI) such that it can be cited independently in the future. For instructions see: http://journals.plos.org/plosone/s/submission-guidelines#loc-laboratory-protocols

We look forward to receiving your revised manuscript.

Kind regards,

Tamar Ringel-Kulka

Academic Editor

PLOS ONE

"This research was funded by Wollo University."

Please remove any funding-related text from the manuscript and let us know how you would like to update your Funding Statement. Currently, your Funding Statement reads as follows:  "No"

3. Thank you for stating the following in your Competing Interests section:  "No"

Reviewers' comments:

Reviewer's Responses to Questions

**Comments to the Author**

1. Is the manuscript technically sound, and do the data support the conclusions?

Reviewer #1: Yes

Reviewer #2: Partly

2. Has the statistical analysis been performed appropriately and rigorously? 

Reviewer #1: Yes

Reviewer #2: No

3. Have the authors made all data underlying the findings in their manuscript fully available?

Reviewer #1: Yes

Reviewer #2: Yes

4. Is the manuscript presented in an intelligible fashion and written in standard English?

Reviewer #1: Yes

Reviewer #2: Yes

5. Review Comments to the Author

Reviewer #1: Review: Determinants of stunting among children aged 6-59 months in Kemissie City Administration, North Eastern Ethiopia: A community-based matched case-control study

Summary: The authors examine stunting in Ethiopia with a case-control method. Find that maternal nutritional status, child’s food priority, duration of breastfeeding, child meat-eating, and diarrheal episodes are associated with stunting.

Recommendation:

Minor revision

General Notes:

• Overall the study does not provide novel insights into the determinants of stunting, but it provides additional evidence of factors that have been previously shown to be associated with stunting. However, the study appears to be well done.

• Eliminate causal language (e.g. abstract, throughout the discussion and conclusion including line 216 “causes of stunting”)

Specific Notes:

Check for typos throughout

Reviewer #2: This manuscript represents a community based individual matched case control study (with

107 cases and 214 controls) where the authors find associations between being stunted among children under 5 hears of age and maternal BMI, food priority, exclusive breast feeding,

consumption of meat and experiencing diarrhea. While the authors have an appropriate and succinct argument for the importance of this study in the introduction, and they also seemed to have responded to the first round of comments, the manuscript requires further revisions to represent an important and novel addition to the literature on child undernutrition.

1. How were data cleaned? Particularly the anthropometric data collection? How were outliers addressed?

2. Were any children missing observations on any of the variables included.

3. Greater description of the explanatory factors including in this study is needed. Such information should not only be relegated to category labels in the tables. Moreover, justification is needed for the specific variables included.

4. Did the authors think about conducting a logistic regression analysis with stunting as the outcome pooling all the data and (as a sensitivity analysis doing a linear regression analysis with the HAZ score as the outcome)? It might make results clearer.

5. Did the authors adjust for the Kebele in the model? Past research has found that variation in undernutrition outcomes can be attributed to higher geographic administration levels.

6. Some English editing is required in the paragraph about the wealth index.

7. What additional factors were included in the regression analyses based on results presented in Table 4? More details about the analysis and the factors included should be part of the table information. For example, were any economic household characteristics or WASH variables included?

8. The authors need to re-structure their results. Cut-down the amount of descriptive statistics presented. Mention a few and then refer readers to those tables. Then, the rest of the results section should focus on estimates from the adjusted regression analyses as those respond to the research questions set up in this manuscript.

9. No new results should be presented in the discussion section. Anything that the authors feel is of importance currently being mentioned in the discussion only for the first time should instead be placed in the results section instead.

10. The first paragraph of the discussion should summarize the study design and the 2-3 main findings and why important/novel. The following paragraphs can be discussion about how the results are similar to or different from past work.

11. The study requires a limitation paragraph within the discussion section.

12. The first sentence of the conclusion needs to be re-written as the authors cannot truly claim that the factors they identified are indeed the main causes of stunting. In addition, they need to localize their findings. Also, the last sentence is too long. Break it up. Consider mentioning specific areas for further research and what are the policy/public health implications of their findings.

6. PLOS authors have the option to publish the peer review history of their article (what does this mean?). If published, this will include your full peer review and any attached files.

Reviewer #1: No

Reviewer #2: No

---

## [Author Response · Author response to Decision Letter 0]

12 Jun 2020

Rebuttal letter

Response to the Journal Requirements Questions 

Question #1: Please ensure that your manuscript meets PLOS ONE's style requirements, including those for file naming.

Response: Thank you for this remark. We re-formatted the revised manuscript using the PLoS ONE format guidelines. The whole content of the manuscript including the abstract, introduction, methods, discussion and reference are seriously formatted using the guidelines (please see the revised version for each section).

Question #2: Thank you for stating the following in the Acknowledgments Section of your manuscript: "This research was funded by Wollo University." We note that you have provided funding information that is not currently declared in your Funding Statement. However, funding information should not appear in the Acknowledgments section or other areas of your manuscript. We will only publish funding information present in the Funding Statement section of the online submission form. Please remove any funding-related text from the manuscript and let us know how you would like to update your Funding Statement. Currently, your Funding Statement reads as follows: "No"

Response: Thank you for this information. We updated accordingly and please see the online information in the editorial manger. Please also see the revised version of the acknowledgment in Lines 321 & 323 from page 15.

Question #3: Thank you for stating the following in your Competing Interests section: "No". Please complete your Competing Interests on the online submission form to state any Competing Interests. If you have no competing interests, please state "The authors have declared that no competing interests exist." as detailed online in our guide for authors at http://journals.plos.org/plosone/s/submit-now. This information should be included in your cover letter; we will change the online submission form on your behalf.

Response: Thank you and we updated based on your suggested. Please see the cover letter. 

Line by line response to reviewers

Reviewer # 1

Reviewer #1: Review: Determinants of stunting among children aged 6-59 months in Kemissie City Administration, North Eastern Ethiopia: A community-based matched case-control study. Summary: The authors examine stunting in Ethiopia with a case-control method. Find that maternal nutritional status, child’s food priority, duration of breastfeeding, child meat-eating, and diarrheal episodes are associated with stunting.

Recommendation:

Minor revision.

General Notes:

.the study appears to be well done.

• Eliminate causal language (e.g. abstract, throughout the discussion and conclusion including line 216 “causes of stunting)

Specific Notes:

Check for typos throughout

Response: Thank you for the positive assessment of our work and your well-articulated feedback is highly appreciated. We revised the manuscript based on your suggestions and any typo error was corrected using a professional language editor. 

Reviewers # 2

Question #2: This manuscript represents a community based individual matched case control study (with 107 cases and 214 controls) where the authors find associations between being stunted among children under 5 years of age and maternal BMI, food priority, exclusive breast feeding, consumption of meat and experiencing diarrhea. While the authors have an appropriate and succinct argument for the importance of this study in the introduction, and they also seemed to have responded to the first round of comments, the manuscript requires further revisions to represent an important and novel addition to the literature on child under nutrition.

Response: Thank you for your kind appreciation for our study and all your comments are well taken and corrected accordingly. Please find for each of them here below. 

Question #1: How were data cleaned? Particularly the anthropometric data collection? How were outliers addressed?

Response: before starting the data collection, data collectors were trained on how to collect data. Coefficient of variation was calculated for each data collector and they were checked whether they are able to collect anthropometric data or not. Close contact supervision was also done by a supervisor and authors. Then, before running the analysis, data were cleaned by running frequency. Based on the WHO Anthro 2005 software, the outlier is detected if the Z score is <-5 or >5. Within this range it is not considered as an outlier. If the Z score value is out of the aforementioned range it is considered as there is an outlier. Outliers were checked by the presence of flag in WHO Anthro 2005 software. The result indicates that there is no outlier (lines 135 to 145). 

Question #2: Were any children missing observations on any of the variables included?

Response: Sorry for the confusion we created, if there is any. But, in this study there is no children with missing observations and unfortunately it is not our concern. Therefore, children with missing variables were not included in the final model. 

Questions #3: Greater description of the explanatory factors including in this study is needed. Such information should not only be relegated to category labels in the tables. Moreover, justification is needed for the specific variables included.

Response: Thank you for this pertinent comment and we updated the manuscript accordingly and we explained the explanatory variable in the result section (lines196-179, 200-204). 

Question #4. Did the authors think about conducting a logistic regression analysis with stunting as the outcome pooling all the data and (as a sensitivity analysis doing a linear regression analysis with the HAZ score as the outcome)? It might make results clearer.

Response: the aim of this was to assess the determinants of stunting using matched case-control. Therefore, the appropriate analysis for the above aim must be a conditional logistic regression since we used a binary outcome of case or control, we did not use any continues outcome. (Please see the data analysis section from lines 146 to 1169)

Question #5. Did the authors adjust for the Kebele in the model? Past research has found that variation in under nutrition outcomes can be attributed to higher geographic administration levels.

Response: Thank you for your feedback and we agree that geographical variation may have impact on under nutrition outcomes. Since the study was conducted among city administrative under five children, initially we did not consider residence (kebeles) as a factor. Therefore, no adjustment was done kebele. Further research in filling this gap is highly recommended. 

Question #6. Some English editing is required in the paragraph about the wealth index.

Response: Thank you for your insight and we updated language editing (please see the revised version lines 152 50 161). 

Question #7. The regression analyses based on results presented in Table 4? More details about the analysis and the factors included should be part of the table information. For example, were any economic household characteristics or WASH variables included? 

Response: Previously we only reported variables who had statistically significant in the final model. According to the comment given by reviewer 2, we included other variables that were entered in to multi-variable model. Water, educational status of mother and wealth index were significant (See Table 4). 

Question #8. The authors need to re-structure their results. Cut-down the amount of descriptive statistics presented. Mention a few and then refer readers to those tables. Then, the rest of the results section should focus on estimates from the adjusted regression analyses as those respond to the research questions set up in this manuscript.

Response: Thank you for this important comments, we tried to update the manuscript. Please see the result section from lines 180 to 232.

Question #9. No new results should be presented in the discussion section. Anything that the authors feel is of importance currently being mentioned in the discussion only for the first time should instead be placed in the results section instead.

Response: We updated the manuscript and please see the first paragraph of the discussion in lines 219 to 232. 

Question #10. The first paragraph of the discussion should summarize the study design and the 2-3 main findings and why important/novel. The following paragraphs can be discussion about how the results are similar to or different from past work.

Response: Please see the first paragraph of the discussion in lines 234 to 240.

Question #11: The study requires a limitation paragraph within the discussion section.

Response: We included the limitation paragraph as suggested (See lines 292 to 294)

Question #12. The first sentence of the conclusion needs to be re-written as the authors cannot truly claim that the factors they identified are indeed the main causes of stunting. In addition, they need to localize their findings. Also, the last sentence is too long. Break it up. Consider mentioning specific areas for further research and what are the policy/public health implications of their findings.

Response: Thank you for this key comment. Please see the updated version of the conclusion in lines 308 to 311.

We would like to thank the reviewers and editors for evaluating our manuscript. We have tried to address all the concerns in a proper way and believe that our paper has improved considerably. We would be happy to make further corrections if necessary and look forward to hearing from you all soon.

---

## [Decision Letter · Decision Letter 1]

3 Sep 2020

Determinants of stunting among children aged 6-59 months in Kemissie City Administration, North Eastern Ethiopia: A matched case-control study

PONE-D-19-23907R1

Dear Dr. Adane (PhD),

We’re pleased to inform you that your manuscript has been judged scientifically suitable for publication and will be formally accepted for publication once it meets all outstanding technical requirements.

Kind regards,

Tamar Ringel-Kulka

Academic Editor

PLOS ONE

Additional Editor Comments (optional):

Reviewers' comments:

Reviewer's Responses to Questions

**Comments to the Author**

1. If the authors have adequately addressed your comments raised in a previous round of review and you feel that this manuscript is now acceptable for publication, you may indicate that here to bypass the “Comments to the Author” section, enter your conflict of interest statement in the “Confidential to Editor” section, and submit your "Accept" recommendation.

Reviewer #1: All comments have been addressed

2. Is the manuscript technically sound, and do the data support the conclusions?

Reviewer #1: Yes

3. Has the statistical analysis been performed appropriately and rigorously? 

Reviewer #1: Yes

4. Have the authors made all data underlying the findings in their manuscript fully available?

Reviewer #1: Yes

5. Is the manuscript presented in an intelligible fashion and written in standard English?

Reviewer #1: Yes

6. Review Comments to the Author

Reviewer #1: (No Response)

7. PLOS authors have the option to publish the peer review history of their article (what does this mean?). If published, this will include your full peer review and any attached files.

Reviewer #1: No

---

## [Editor Report · Acceptance letter]

16 Sep 2020

PONE-D-19-23907R1 

Priorities for intervention of childhood stunting in northeastern Ethiopia:                                                              A matched case-control study 

Dear Dr. Adane (PhD):

I'm pleased to inform you that your manuscript has been deemed suitable for publication in PLOS ONE. Congratulations! Your manuscript is now with our production department. 

Kind regards, 

on behalf of

Dr. Tamar Ringel-Kulka 

Academic Editor

PLOS ONE